# A Novel Wide-Band Directional MUSIC Algorithm Using the Strength Proportion

**DOI:** 10.3390/s23094562

**Published:** 2023-05-08

**Authors:** Wencong Xu, Bingshu Chen, Yue Hu, Jianxun Li

**Affiliations:** 1Department of Automation, Shanghai Jiao Tong University, No. 800 Road Dongchuan, Shanghai 200240, China; 2Department of Electrical Engineering, Shanghai Jiao Tong University, No. 800 Road Dongchuan, Shanghai 200240, China

**Keywords:** partial discharge location, wide-band, strength proportion

## Abstract

The directional multiple signal classification (Dir-MUSIC) algorithm based on the antenna gain array manifold has been proposed to find the direction of the partial discharge (PD) source in substations. However, PD signals are wideband signals and the antenna gain pattern functions are always different at different frequencies; therefore, the accuracy can be improved using a wideband Dir-MUSIC algorithm. In this paper, wideband Dir-MUSIC algorithms are discussed and a novel wideband Dir-MUSIC algorithm using the strength proportion (DirSP) is proposed. This algorithm estimates a focusing PD signal at a certain frequency using the strength proportion among different directions, and then the Dir-MUSIC algorithm can process the focusing PD signal at this frequency. In simulations, when the antenna gain functions among different frequency bins are quite different, the Dir-MUSIC algorithm loses accuracy; meanwhile, DirDP performs very well. In the experiments, we deal with six sets of samples, and the mean error and standard deviation are both smaller than 4° better than other methods.

## 1. Introduction

Partial discharge (PD) detection and location systems are indispensable for performing quality assurance and fault identification in high-voltage apparatus. PD can be detected and located through PD signals (electromagnetic waves), especially the ultra-high-frequency (UHF) component of PD signals [1,2,3]. In recent years, PD detection and location methods based on UHF signals have been popular, with the advantages of strong anti-interference, high sensitivity and a stable transmitting speed. Thus, UHF detection methods, especially those using an omnidirectional UHF sensor array to detect and locate the PD signals in substations, have obtained fruitful results [4,5,6,7,8,9].

In some situations, the omnidirectional UHF sensor array is not a good choice because it need a small and light sensor array. For example, the intelligent inspection robot used for the substation inspection system can only carry a small sensor array [10,11]. However, PD detection methods using the time difference of arrival (TDOA) algorithm based on an omnidirectional UHF sensor array require a relatively large size sensor array to obtain an accurate TDOA. Therefore, a directional antenna array is a new choice in this situation. As a tentative research direction, the directional multiple signal classification (Dir-MUSIC) algorithm for direction of arrival (DOA) estimation based on signal strength, as represented by antenna gain array manifold, is proposed [12]. A miniaturized spiral directional antenna array has been designed and developed to receive the PD signal. Using the signal strength information, the Dir-MUSIC algorithm can successfully and precisely find the direction of the PD source. However, in the Dir-MUSIC algorithm, the antenna gain is assumed to be approximatively equal at different frequencies of interest. In fact, when the antenna gain at different frequencies is too different, the direction error will be unacceptable. Therefore, it is necessary to improve a wideband Dir-MUSIC algorithm.

It is difficult to deal with the wideband signals directly using a subspace DOA estimation algorithm. Most wideband subspace methods decompose the sensor outputs into several narrowband signals using a filter bank or the discrete Fourier transform (DFT). Normally, they nclude the incoherent signal subspace method (ISSM) [13] and the coherent signal subspace method (CSSM) [14]. ISSM is one of the simplest wideband methods. It process the decomposed narrowband signals independently, applying narrowband techniques, and the results of all the frequency bins are averaged to obtain the final DOA estimation. CSSM calculates a focusing matrix between different frequency bins and estimates a coherently averaged sample covariance matrix, which can be obtained by applying narrowband techniques. CSSM requires initial values to find the focusing matrices, and the estimation performance of CSSM is sensitive to these initial values.

Wideband subspace techniques have many applications. An efficient implementation of CSSM on the field-programmable gate array (FPGA) has been proposed to achieve the desired performance [15]. A generic subspace model has been proposed for characterizing a variety of radio frequency interference (RFI) types, which reveals a low-dimensional structure of the RFI subspace [16]. For PD location in substations in oil, a PD location method based on an ultrasonic phased array and wideband array signal processing is proposed [17].

In this paper, based on the Dir-MUSIC algorithm, we propose a novel wideband Dir-MUSIC algorithm using the strength proportion (DirSP), which can be regarded as another form of CSSM. In the proposed algorithm, PD signals will be decomposed into several narrowband signals using the DFT. We will choose some frequency bins to estimate an averaged sample covariance matrix at a certain frequency, and we will first estimate a focusing PD signal at this frequency. Then, we could apply the Dir-MUSIC algorithm on the averaged sample covariance matrix and the focusing PD signal.

## 2. Wide-Band Dir-MUSIC Algorithms

A PD signal is always a microsecond pulse signal, and the time interval between two pulses occurs randomly at the interval 50 us, 1 ms. Therefore, the probability of multi-source signal pulse superposition is very small, and this paper can only consider a single PD source in one time. We consider a uniform circular array of *N* elements (directional antenna), and the opening direction of each element is toward outside. The bandwidths of PD signals need not be identical, but there should be some frequency band $\left[\omega_{L},\omega_{H}\right]$ where PD signals’ bandwidths overlap.

Most wideband subspace methods decompose the sensor outputs into several narrowband signals using a filter bank or the DFT. Generally, PD signals are always measured at some frequency band of interest, such as the designed resonance points of antennas. If $\left[\omega_{L},\omega_{H}\right]$ contains all the frequency bands of interest, then the output of the filter bank or DFT module can be written in vector form at *K* frequencies, as follows:(1)$$X\left(\omega_{i}\right)=G\left(\omega_{i},\theta \right)S\left(\omega_{i}\right)+N\left(\omega_{i}\right),i=1,\cdots ,K,$$
where
(2)$$X\left(\omega_{i}\right)={\left[X_{1}\left(\omega_{i}\right),\cdots ,X_{N}\left(\omega_{i}\right)\right]}^{T},$$
where $X_{j}\left(\omega_{i}\right)$ is the output of the *j*th antenna at frequency $\omega_{i}$ for j=1,⋯,N, i=1,⋯,K and $\omega_{L}<\omega_{i}<\omega_{H}$. $S(\omega_{i})$ is the component of the PD source and $G(\omega_{i},\theta )$ is the antenna gain matrix at frequency $\omega_{i}$ ($G(\omega_{i},\theta )$ is a vector in this paper):(3)$$G\left(\omega_{i},\theta \right)={\left[g_{1}\left(\omega_{i},\theta_{1}\right),\cdots ,g_{N}\left(\omega_{i},\theta_{N}\right)\right]}^{T}.$$
where $g_{j}\left(\omega_{i},\theta_{j}\right)$ is the *j*th antenna gain when the direction angle of the PD source is $\theta_{j}$ at frequency $\omega_{i}$, and $\theta_{j+1}=\theta_{j}+\frac{360}{N}$ holds.

### 2.1. The Normal Incoherent and Coherent Method

Next, a normal incoherent method is introduced. First, find the direction using the Dir-MUSIC algorithm at frequency $\omega_{i}$, as follows:(4)$$\theta_{i}=\mathrm{Dir-MUSIC}\left(X\left(\omega_{i}\right),G\left(\omega_{i},\theta \right)\right).$$

Then take an arithmetic mean of $\theta_{i}$ as the estimated direction:(5)$$\theta_{est}=\sum_{i=1}^{K}\theta_{i}.$$

In the coherent method, an estimation of the coherently averaged sample covariance matrix is indispensable. Focusing matrices are effective in this operation, and can transform the constituent narrow-band components of the received signal into appropriate narrow-band representations at a certain given frequency. Next, we will introduce the rotational signal subspace (RSS) focusing matrix $T(\omega_{i})$, which satisfies the following constrained minimization problem:(6)$$\min_{T(\omega_{i},\theta )}{∥G\left(\omega_{1},\theta \right)-T\left(\omega_{i},\theta \right)G\left(\omega_{i},\theta \right)∥}_{F},i=1,\cdots ,K,$$
subject to
$$T^{T}\left(\omega_{i},\theta \right)T\left(\omega_{i},\theta \right)=I,$$
where ${∥∥}_{F}$ is the Frobenius matrix norm, and $\omega_{1}$ is the centre frequency. One solution to (Equation 6) is
(7)$$T\left(\omega_{i},\theta \right)=V\left(\omega_{i},\theta \right)U{\left(\omega_{1},\theta \right)}^{T},$$
where the columns of $U(\omega_{1},\theta )$ and $V(\omega_{i},\theta )$ are the left and right singular vectors of $G\left(\omega_{i},\theta \right)G{\left(\omega_{1},\theta \right)}^{T}$.

Then, the coherently averaged sample covariance matrix at $\omega_{1}$ can be estimated by
(8)$$R_{1}=\left(\sum_{i=1}^{K}T\left(\omega_{i}\right)X\left(\omega_{i}\right)\right){\left(\sum_{i=1}^{K}T\left(\omega_{i}\right)X\left(\omega_{i}\right)\right)}^{T}.$$

It must be noted that directly applying the RSS focussing matrix is not a good choice. $G(\omega_{1},\theta )$ and $G(\omega_{i},\theta )$ in (Equation 6) always have different modules, so *T* can not have a theoretical solution that satisfies $∥G\left(\omega_{1},\theta \right)-T\left(\omega_{i},\theta \right)G\left(\omega_{i},\theta \right){∥}_{F}=0$. In the traditional CSSM, the steering vectors have the same modules, and this characteristic ensures the effectiveness of the traditional CSSM. Therefore, when the antenna gain vectors have quite different modules, CSSM in this paper will lose accuracy.

As an alternative, a wideband Dir-MUSIC algorithm using the strength proportion is proposed in the next subsection.

### 2.2. The Wide-Band Dir-MUSIC Algorithm Using the Strength Proportion

In this paper, $g_{j}\left(\omega_{i},\theta \right)$ is only a real number when $\omega_{i},\theta$ is given. Then, from (Equation 1) and (Equation 3), we can obtain
(9)$$\begin{array}{c} \begin{array}{cc} X_{j}\left(\omega_{1}\right) & =g_{j}\left(\omega_{1},\theta \right)S\left(\omega_{1}\right)+N_{j}\left(\omega_{1}\right)\\ X_{j}\left(\omega_{2}\right) & =g_{j}\left(\omega_{2},\theta \right)S\left(\omega_{2}\right)+N_{j}\left(\omega_{2}\right)\\ \; & \cdots \\ X_{j}\left(\omega_{K}\right) & =g_{j}\left(\omega_{K},\theta \right)S\left(\omega_{K}\right)+N_{j}\left(\omega_{K}\right)\\ j & =1,\cdots ,N. \end{array} \end{array}$$

Let $\omega_{1}$ be the reference frequency; (Equation 9) could be written as
(10)$$\begin{array}{c} \begin{array}{cc} X_{j}\left(\omega_{1}\right) & =g_{j}\left(\omega_{1},\theta \right)S\left(\omega_{1}\right)+N_{j}\left(\omega_{1}\right)\\ \frac{g_{j}\left(\omega_{1},\theta \right)}{g_{j}\left(\omega_{2},\theta \right)}X_{j}\left(\omega_{2}\right) & =g_{j}\left(\omega_{1},\theta \right)S\left(\omega_{2}\right)+\frac{g_{j}\left(\omega_{1},\theta \right)}{g_{j}\left(\omega_{2},\theta \right)}N_{j}\left(\omega_{2}\right)\\ \; & \cdots \\ \frac{g_{j}\left(\omega_{1},\theta \right)}{g_{j}\left(\omega_{K},\theta \right)}X_{j}\left(\omega_{K}\right) & =g_{j}\left(\omega_{1},\theta \right)S\left(\omega_{K}\right)+\frac{g_{j}\left(\omega_{1},\theta \right)}{g_{j}\left(\omega_{K},\theta \right)}N_{j}\left(\omega_{K}\right)\\ j & =1,\cdots ,N. \end{array} \end{array}$$

Let ${\tilde{X}}_{j}$ be the sum of the left side in (Equation 10):(11)$${\tilde{X}}_{j}=\sum_{i=1}^{K}\frac{g_{j}\left(\omega_{1},\theta \right)}{g_{j}\left(\omega_{i},\theta \right)}X_{j}\left(\omega_{i}\right)=g_{j}\left(\omega_{1},\theta \right)S+\sum_{i=1}^{K}\frac{g_{j}\left(\omega_{1},\theta \right)}{g_{j}\left(\omega_{i},\theta \right)}N_{j}\left(\omega_{i}\right).$$

Then, the coherently averaged sample covariance matrix at $\omega_{1}$ can be estimated by
(12)$$\begin{array}{c} R_{1}=\tilde{X}{\tilde{X}}^{T}, \end{array}$$
where $\tilde{X}={\left[{\tilde{X}}_{1},\cdots ,{\tilde{X}}_{N}\right]}^{T}$.

Next, we will explain that this method is equivalent to CSSM when it is applied in the normal array signal processing.

We consider a single source and a linear array, and the incident angle is θ. Then (Equation 9) and (Equation 10) convert to
(13)$$\begin{array}{c} \begin{array}{cc} X_{j}\left(\omega_{1}\right) & =e^{-i\omega_{1}d_{j}\sin\theta }S\left(\omega_{1}\right)+N_{j}\left(\omega_{1}\right)\\ X_{j}\left(\omega_{2}\right) & =e^{-i\omega_{2}d_{j}\sin\theta }S\left(\omega_{2}\right)+N_{j}\left(\omega_{2}\right)\\ \; & \cdots \\ X_{j}\left(\omega_{K}\right) & =e^{-i\omega_{K}d_{j}\sin\theta }S\left(\omega_{K}\right)+N_{j}\left(\omega_{K}\right)\\ j & =1,\cdots ,N. \end{array} \end{array}$$
(14)$$\begin{array}{c} \begin{array}{cc} X_{j}\left(\omega_{1}\right) & =e^{-i\omega_{1}d_{j}\sin\theta }S\left(\omega_{1}\right)+N_{j}\left(\omega_{1}\right)\\ e^{-i\left(\omega_{1}-\omega_{2}\right)d_{j}\sin\theta }X_{j}\left(\omega_{2}\right) & =e^{-i\omega_{1}d_{j}\sin\theta }S\left(\omega_{2}\right)\\ & +e^{-i\left(\omega_{1}-\omega_{2}\right)d_{j}\sin\theta }N_{j}\left(\omega_{2}\right)\\ \; & \cdots \\ e^{-i\left(\omega_{1}-\omega_{N}\right)d_{j}\sin\theta }X_{j}\left(\omega_{2}\right) & =e^{-i\omega_{1}d_{j}\sin\theta }S\left(\omega_{2}\right)\\ & +e^{-i\left(\omega_{1}-\omega_{N}\right)d_{j}\sin\theta }N_{j}\left(\omega_{N}\right)\\ j & =1,\cdots ,N. \end{array} \end{array}$$

Let
(15)$$T\left(\omega_{i}\right)=diag\left\{1,e^{-i\left(\omega_{1}-\omega_{2}\right)d_{j}\sin\theta },\cdots ,e^{-i\left(\omega_{1}-\omega_{N}\right)d_{j}\sin\theta }\right\}.$$

The coherently averaged sample covariance matrix of this method and the CSSM are identical. Therefore, the proposed method is an application of CSSM in this paper.

Now, we can estimate the direction of the PD source using the Dir-MUSIC algorithm, and the steps are as follows.

We calculate the eigenvalues and eigenvectors of $R_{1}$ via eigenvalue decomposition. It is clear that $R_{1}$ is a real symmetric matrix, and the eigenvalues are all real numbers. $Span\{R_{1}\}$ also consists of signal subspace $Span\{R_{S}\}$ and noise subspace $Span\{R_{N}\}$:(16)$$R_{1}=G_{1}R_{S}G_{1}^{T}+R_{N},$$
where $G_{1}$ is a simple expression of $G(\omega_{1},\theta )$. Therefore, $R_{1}$ is a positive definite. Since $\lambda_{1}\geq \lambda_{2}\geq \cdots \geq \lambda_{N}$ are the eigenvalues, and $\nu_{1},\nu_{2},\cdots ,\nu_{N}$ are the related corresponding eigenvectors. Additionally, the eigenvectors are orthogonal to each other; that is,
(17)$$\nu_{i_{1}}^{T}\nu_{i_{2}}=0,1\leq i_{1}\neq i_{2}\leq N.$$

This paper considers only one signal at one time; therefore, there is only one eigenvalue related to the signal. Certainly, $\lambda_{1}$ is the biggest eigenvalue related to the signal, and suppose $\lambda_{i}$ is one of the smaller eigenvalues. Then, we have
(18)$$\begin{array}{c} \begin{array}{cc} R_{1}\nu_{i}= & \;\lambda_{i}\nu_{i}\\ \left(G_{1}R_{S}G_{1}^{T}+R_{N}\right)\nu_{i}= & \;\lambda_{i}\nu_{i}\\ \left(G_{1}R_{S}G_{1}^{T}+\sigma^{2}I\right)\nu_{i}= & \;\lambda_{i}\nu_{i}\\ G_{1}R_{S}G_{1}^{T}\nu_{i}= & \;1. \end{array} \end{array}$$

Because $R_{S}$ is positive definite, we have
(19)$$G_{1}^{T}\nu_{i}=0,i=2,\cdots ,N.$$

From (Equation 3), we can see that $G_{1}$ is also a vector function of θ; therefore, we can search the direction of the PD source by traversing the possible value of θ. Let N−1 eigenvectors be related to the smaller N−1 eigenvalue to construct a noise matrix:(20)$$E_{n}=\left[\nu_{2},\nu_{3},\cdots ,\nu_{N}\right].$$

Let the space spectrum be:(21)$$P_{mu}\left(\theta \right)=\frac{1}{g_{1}^{T}\left(\theta \right)E_{n}E_{n}^{T}g_{1}\left(\theta \right)}=\frac{1}{∥E_{n}^{T}g_{1}\left(\theta \right)∥},$$
where $g_{1}\left(\theta \right)$ represents $G_{1}$.

When we take the maximum value of $P_{mu}\left(\theta \right)$, it indicates that $∥E_{n}^{T}g_{1}\left(\theta \right)∥$ is closest to 0 in all θ; then, we decide this θ is the estimated direction angle.

## 3. Simulations

In this section, simulations will be carried out to verify the advantages of the wideband Dir-MUSIC algorithm. A relatively simple but effective wideband PD signal is designed for simulations.

A double exponential oscillation attenuation function is used to simulate the PD signal [18], and the specific expression is
(22)$$f\left(t\right)=\left(\exp\left(-\frac{1.3t}{k_{1}}\right)-\exp\left(-\frac{2.2t}{k_{2}}\right)\right)sin\left(2\pi t_{f}\right),$$
where the parameters can be $k_{1}=1.5\times 10^{-8},k_{2}=2.0\times 10^{-8},f=1.0\times 10^{8}$. This is a narrow-band signal in the approximate frequency band [90M,110M]. Consider other sets of parameters $k_{1}=1.5\times 10^{-8},k_{2}=2.0\times 10^{-8},f=2.0\times 10^{8}$ and $k_{1}=1.5\times 10^{-8},k_{2}=2.0\times 10^{-8},f=3.0\times 10^{8}$; the related simulated PD signals are written as $f_{2}\left(t\right)$ and $f_{3}\left(t\right)$, and their approximate frequency bands are [190M,210M] and [290M,310M]. Meanwhile, the simulated PD signal related to the first set of parameters is written as $f_{1}\left(t\right)$. Then, a wide-band PD signal is simulated as
(23)$$f\left(t\right)=\frac{1}{3}\left(f_{1}\left(t\right)+f_{2}\left(t\right)+f_{3}\left(t\right)\right).$$

The sampled pulse signal of f(t) without noise at the sample rate $1\times e^{-9}$ is shown in Figure 1, and the frequency spectrum obtained from DFT is shown in Figure 2.

From Figure 1 and Figure 2, this simulated PD signal is a wideband signal that contains three bands. $\omega_{1}=100M$, $\omega_{2}=200M$, $\omega_{3}=300M$ are the centre frequencies of the three bands. When the antenna gain is different among $\omega_{1}$, $\omega_{2}$ and $\omega_{3}$, the Dir-MUSIC algorithm will lose its effectiveness.

Based on the actual antenna pattern, a linear combination of three Gaussian functions is enough to describe the two-dimensional antenna pattern function, as shown in
(24)$$g\left(\theta \right)=a_{1}e^{-\frac{{\left(\theta -b_{1}\right)}^{2}}{c_{1}^{2}}}+a_{2}e^{-\frac{{\left(\theta -b_{2}\right)}^{2}}{c_{2}^{2}}}+a_{3}e^{-\frac{{\left(\theta -b_{3}\right)}^{2}}{c_{3}^{2}}}.$$

Suppose the set of parameters at $\omega_{1}$ is $a_{1}=0.5255,\;b_{1}=218.1,\;c_{1}=51.73,$ $a_{2}=0.3405,\;b_{2}=304.5,\;c_{2}=41,\;a_{3}=0.6251,\;b_{3}=156.1,\;c_{3}=109.1$, and the other two set sof parameters are different at $c_{1}=101.73$ and $c_{1}=251.73$. The normalized antenna gains of the three bands are shown in Figure 3.

We provide simulation steps in Figure 4.

Let the direction of the PD source θ change from 91° to 270°, and SNR is set at 5, 0, −5, and the direction results of DirSP compared to Dir-MUSIC are shown in Figure 5, Figure 6 and Figure 7.

From Figure 5, Figure 6 and Figure 7, we can see when the antenna pattern functions among different frequency bins are quite different; the Dir-MUSIC algorithm only using one function will lose accuracy. Meanwhile, the DirSP algorithm using three functions performs very well.

Let the direction of the PD source θ change from 91° to 270°; SNR is set at 5, 0, −5, and the direction results of the wide-band methods are shown in Figure 8, Figure 9 and Figure 10. Additionally, each result in the following figures is a mean of 50 random estimations.

From Figure 8, Figure 9 and Figure 10, we can see that the errors of ISSM and DirSP are similar, but much less than CSSM. Specifically, the mean errors of ISSM, CSSM and DirSP are 1.8130°, 5.1066° and 2.1403° at SNR =−5, 0.9914°, 2.7778° and 1.1081° at SNR =0, 0.5532°, 1.4948° and 0.5677° at SNR =5. This is ineffective for CSSM when directly applied in the wide-band Dir-MUSIC algorithm. On the other hand, errors in ISSM and DirSP are acceptable even when SNR is −5.

It seems that ISSM is a little better than the proposed DirSP. The reason for this is that measurements are generated from the ideal model perfectly, and every Dir-MUSIC estimation is a perfect estimation of ISSM. Additionally, in DirSP, the strength proportion in (Equation 10) is always not equal to 1, and the noise distribution changes in $\tilde{X}$ in (Equation 12). Therefore, ISSM performs best in the ideal model. However, the relationship between the actual measurements and the measured antenna pattern are not as ideal. Moreover, the proportion of $f_{1},f_{2}$ and $f_{3}$ is always not equal in (Equation 23); having the wrong proportion will cause mistakes in direction estimation. Next, we simulate a new PD signal as
(25)$$f\left(t\right)=\frac{1}{3}\left(f_{1}\left(t\right)+2f_{2}\left(t\right)+3f_{3}\left(t\right)\right).$$

Then, we repeat the last simulation with the new signal, and the direction results are shown in Figure 11, Figure 12 and Figure 13.

From Figure 11, Figure 12 and Figure 13, we can see that ISSM errors become higher when the proportion of frequency bins is wrong. Specifically, the mean errors of ISSM, CSSM and DirSP are 3.4939°, 3.7559° and 2.6799° at SNR =−5, 1.6623°, 2.1102° and 1.3461° at SNR =0, 0.8794°, 1.1434° and 0.7001° at SNR =5. Another method could possibly be used to estimate the proportion; however, this will increase computational complexity and the accuracy will rely on the proportion estimation. Fortunately, errors in the new method are not influenced with changes in proportion and the errors of DirSP are quite acceptable.

At last, we should analyze why CSSM performs badly in the simulations. $\omega_{0}$ and $\omega_{i}$ in (Equation 6) always have different modules, so *T* could not have a theoretical solution. In the traditional CSSM, the steering vectors have the same modules, and this characteristic ensures the effectiveness of the traditional CSSM. Therefore, when the antenna gain vectors have quite different modules, CSSM in this paper will lose accuracy.

## 4. Experimental Data Processing

Experiments were carried out to verify the effectiveness of DirSP. The experimental platform consisted of a digital oscilloscope with storage function, four equal-length radio frequency coaxial cables, the developed uniform fan-shaped directional Vivaldi antenna array, and a lighter as a signal generator. Figure 14 and Figure 15 are two photos of the experiment environment.

To achieve a good direction, high gain and high angular resolution requirements, we used the developed uniform fan-shaped directional Vivaldi antenna array to receive signals. These four antennas were denoted from Antenna 1 to Antenna 4 in counter-clockwise order. The angle between the symmetry axes of Antenna 1 and Antenna 4 was 60°, and the angle between the symmetry axes of adjacent antennas was 20°. The location of the experimental site was calibrated by infrared range finders, and PD signals were generated by the lighter. We provide an example of the PD signal received from the lighter.

In Figure 16, the signal contains 50 pulses. We used a high sample rate of 10 GHz to verify our algorithm. We calculated the frequency spectrum of a single pulse using DFT, as shown in Figure 17. Normally, interference is stronger at low frequencies than at high frequencies. In this paper, we chose the frequency band 1–2 GHz for performance verification. Whether the frequency band 0.5–1 GHz can be used for the algorithms could be studied in subsequent research.

From Figure 18, we chose three frequency bands, 1.3–1.5 GHz, 1.5–1.7 GHz and 1.7–1.9 GHz, to run the wideband Dir-MUSIC algorithm. We measured the antenna patterns for all four antennas every 50M in the band 1–2 GHz. Because it is a fan-shaped array, we only measured the antenna pattern every 0.5°, from −90° to 90°, and the 0° direction was the direction of the symmetry axis. The measured normalized patterns at 1.4 GHz, 1.6 GHz and 1.8 GHz of all four antennas are shown in Figure 18 and Figure 19.

Next, we will process the experimental data with the wideband Dir-MUSIC algorithm, and the mean error and standard deviation of the direction results will be shown in Table 1. Additionally, it should be noted that the location area is the fan-shaped area, which means that the true azimuth is in [0°,60°].

In Table 1, the first column is the PD coordinate (distance and azimuth). At each position, angles in the first row are mean errors (between the average direction result and the true azimuth) and angles in the second row are standard deviation. The mean errors of DirSP are smaller than other methods in most situations. Only at (7.8 m, 30°) are the mean errors of ISSM and CSSM smaller than DirSP. The standard deviation of DirSP is also smaller than other methods in most situations. More importantly, the mean error and standard deviation are both smaller than 4°. Bad results can be obtained using other methods. In general, the experimental data-processing results are consistent with simulations and DirSP performs better than other methods.

## 5. Conclusions

In this paper, wideband Dir-MUSIC algorithms to find PD directions are discussed, and a novel wideband Dir-MUSIC algorithm using the strength proportion is proposed. Conclusions are drawn as follows.

When the antenna gain pattern functions among different frequency bins are quite different, the Dir-MUSIC algorithm only using one function will lose accuracy. Meanwhile, DirSP using three functions performs very well.

It is non-effective for CSSM to be directly applied to the Dir-MUSIC algorithm because the steer vectors have different modules, and DirSP is another application of CSSM in this paper.

ISSM may perform better when the signals are generated in an ideal model. However, DirSP performs better when the strength proportion among the frequency bins is unknown. DirSP is widely applicable in most situations.

Wideband Dir-MUSIC algorithms can avoid the influence of interference signals.

Finally, experimental data-processing has verified the effectiveness of DirSP. Compared to Dir-MUSIC, ISSM and CSSM, DirSP performs better in terms of both mean errors and standard deviation. More importantly, DirSP is more stable than other methods, while some bad results are obtained using other methods. Therefore, DirSP is effective for PD location.

## Figures and Tables

**Figure 1 sensors-23-04562-f001:**
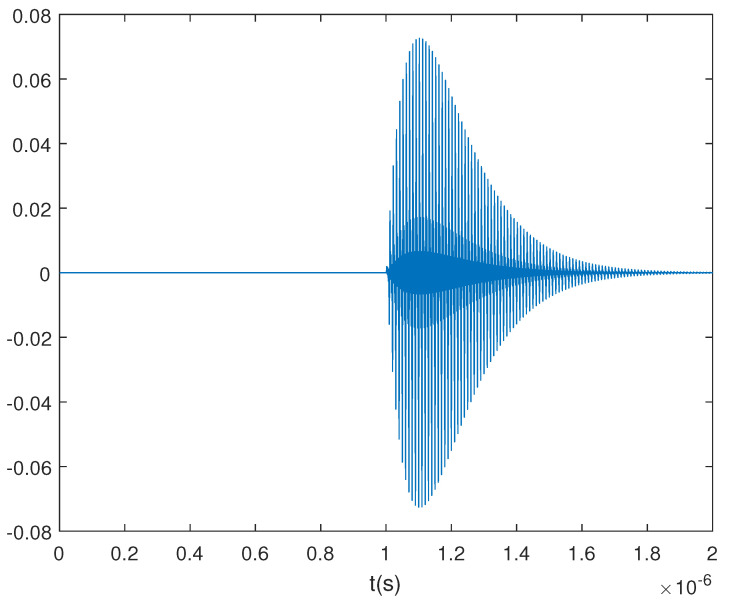
The sampled pulse signal.

**Figure 2 sensors-23-04562-f002:**
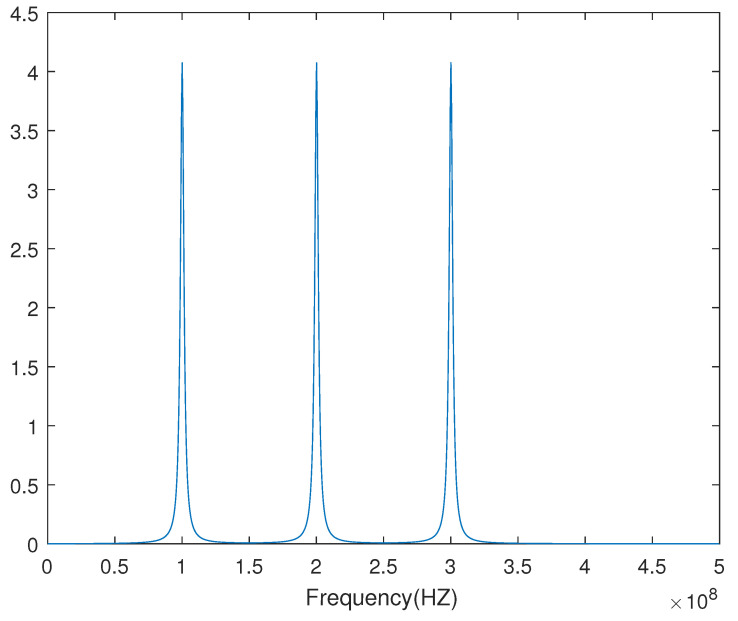
The frequency spectrum of the sampled pulse signal.

**Figure 3 sensors-23-04562-f003:**
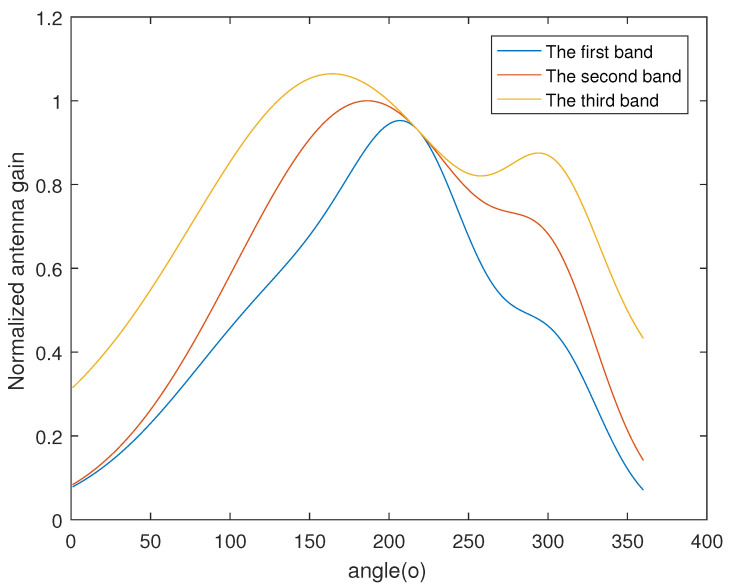
The normalized antenna gain of three bands.

**Figure 4 sensors-23-04562-f004:**
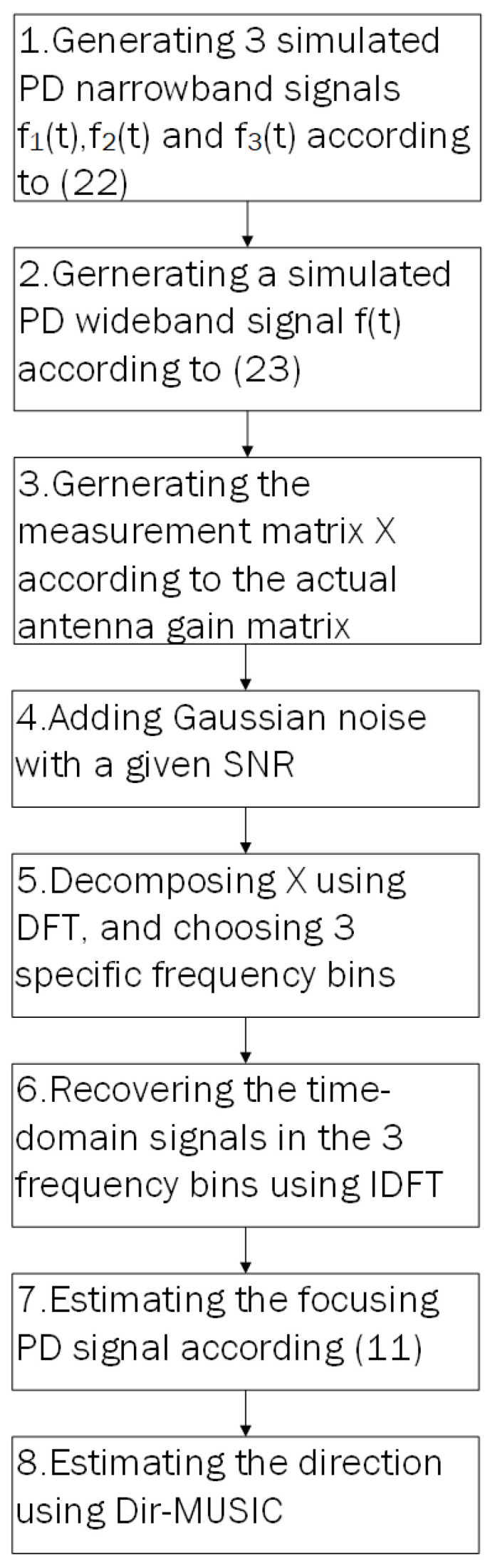
Simulation steps of DirSP.

**Figure 5 sensors-23-04562-f005:**
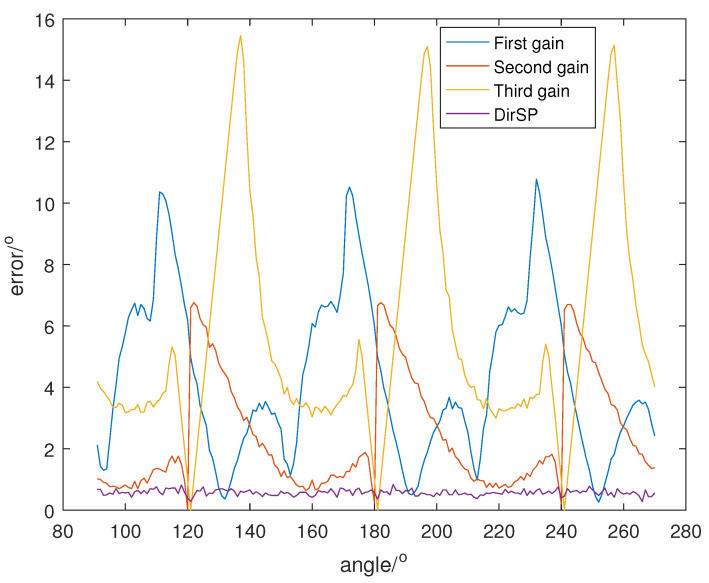
Direction errors of DirSP compared to Dir-MUSIC at SNR = 5.

**Figure 6 sensors-23-04562-f006:**
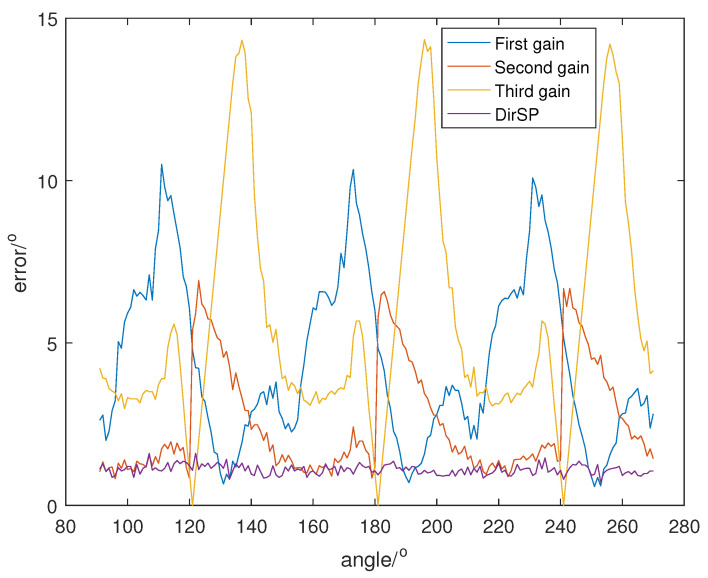
Direction errors of DirSP compared to Dir-MUSIC at SNR = 0.

**Figure 7 sensors-23-04562-f007:**
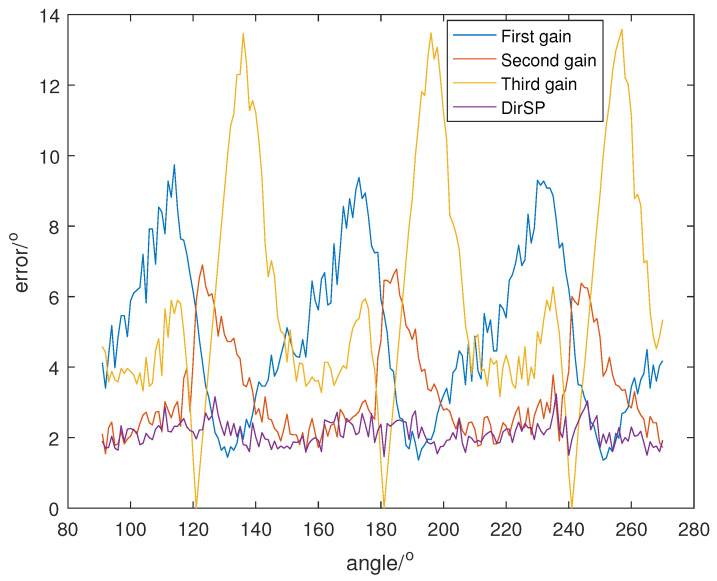
Direction errors of DirSP compared to Dir-MUSIC at SNR = −5.

**Figure 8 sensors-23-04562-f008:**
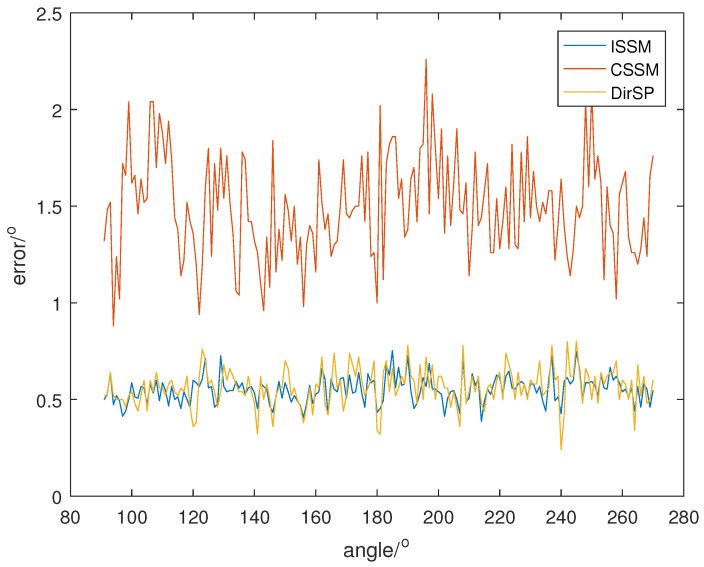
Direction errors of wide-band algorithms at SNR =5.

**Figure 9 sensors-23-04562-f009:**
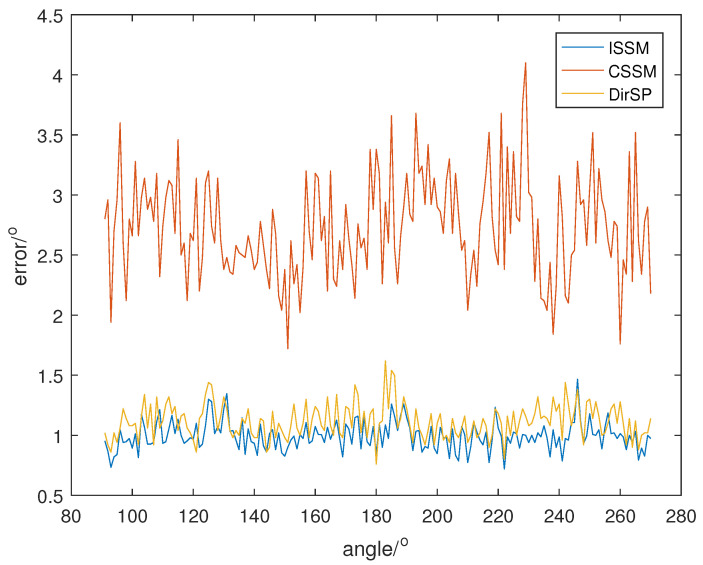
Direction errors of wide-band algorithms at SNR =0.

**Figure 10 sensors-23-04562-f010:**
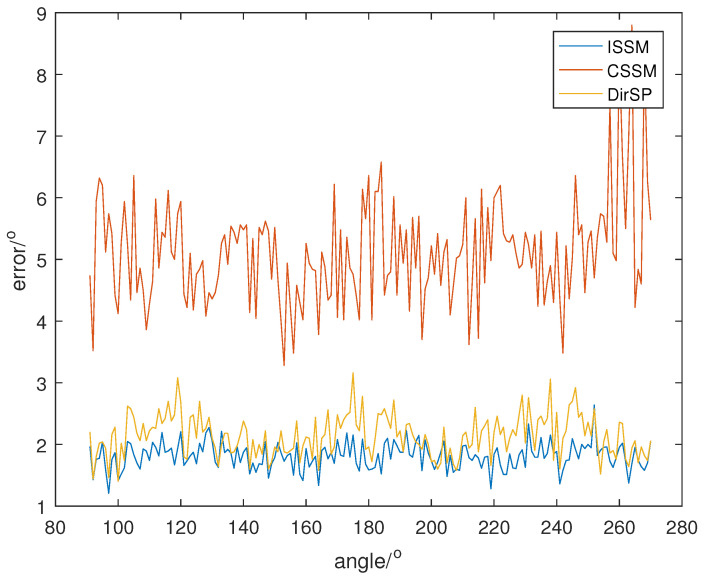
Direction errors of wide-band algorithms at SNR =−5.

**Figure 11 sensors-23-04562-f011:**
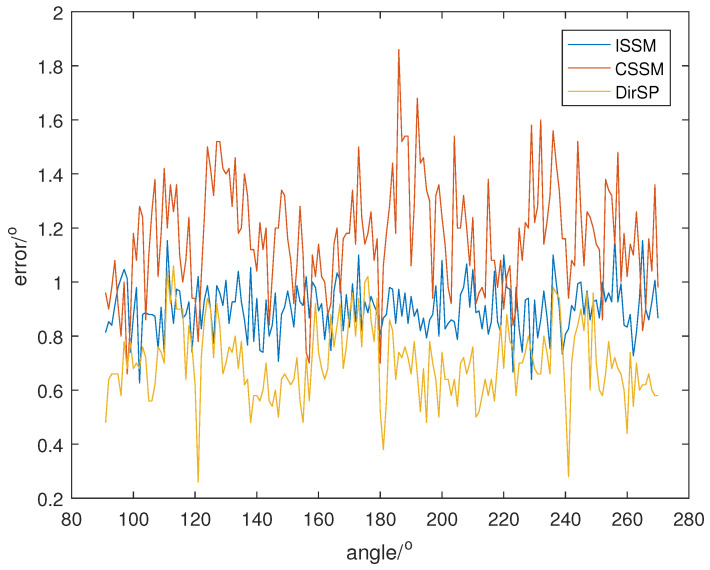
Direction errors of wide-band algorithms at SNR =5 with a new signal.

**Figure 12 sensors-23-04562-f012:**
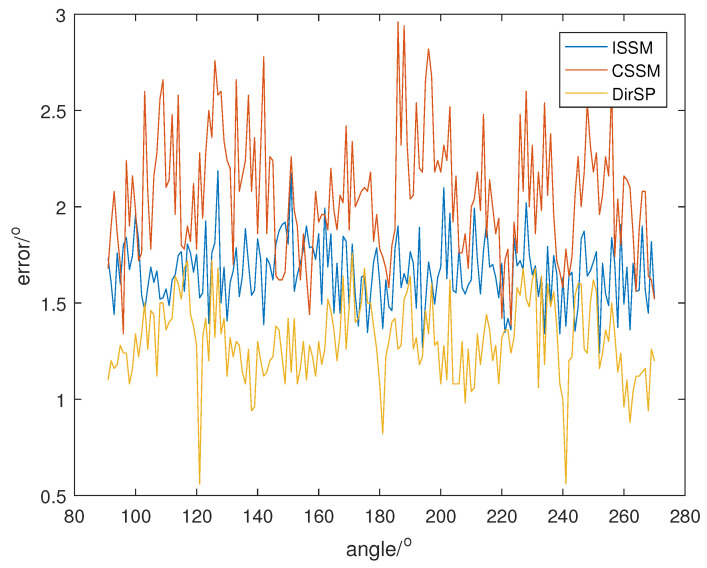
Direction errors of wide-band algorithms at SNR =0 with a new signal.

**Figure 13 sensors-23-04562-f013:**
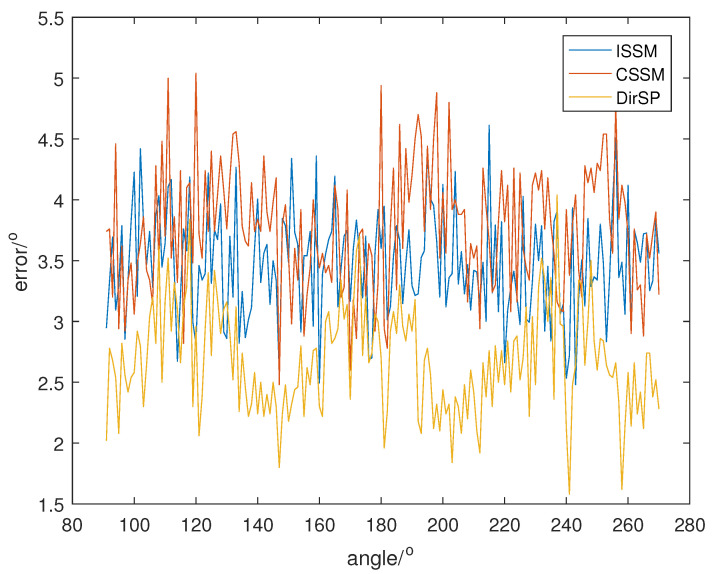
Direction errors of wide-band algorithms at SNR =−5 with a new signal.

**Figure 14 sensors-23-04562-f014:**
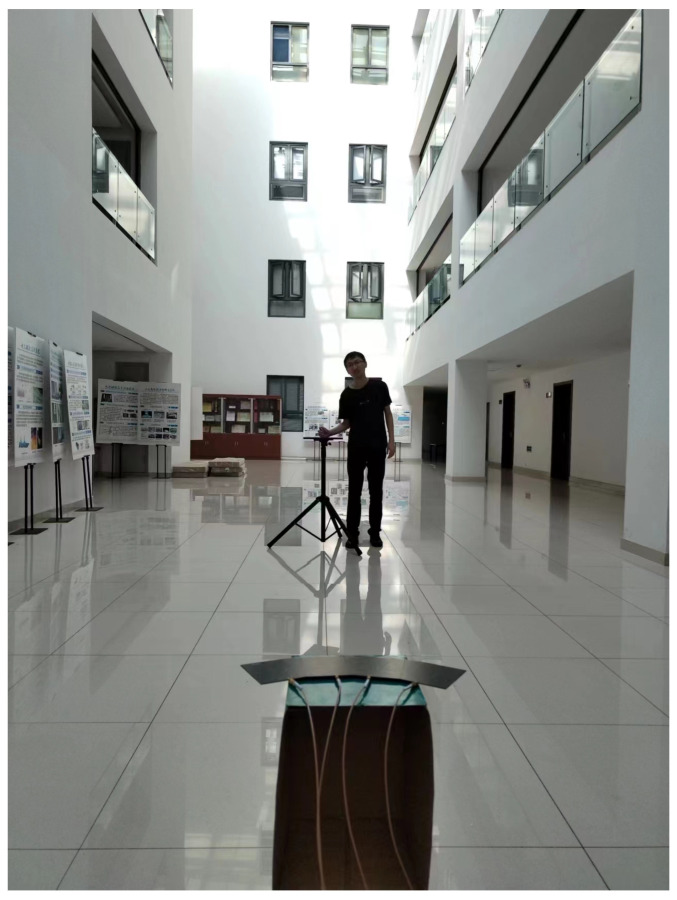
A photo of experiment environment.

**Figure 15 sensors-23-04562-f015:**
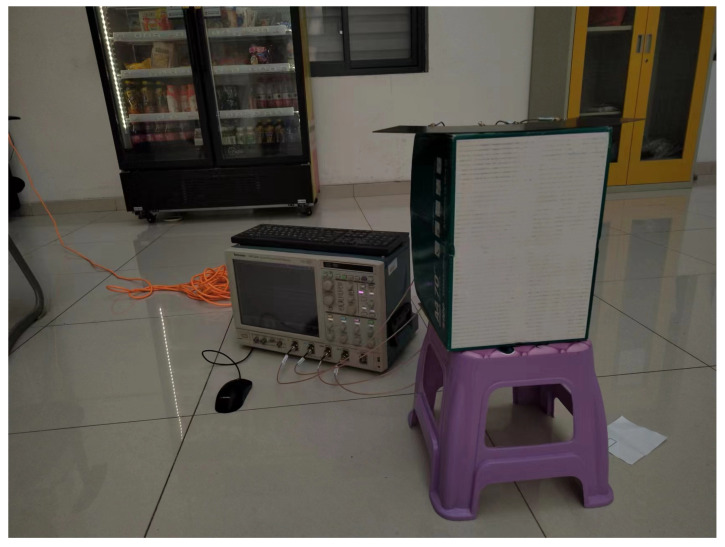
A photo of experiment environment.

**Figure 16 sensors-23-04562-f016:**
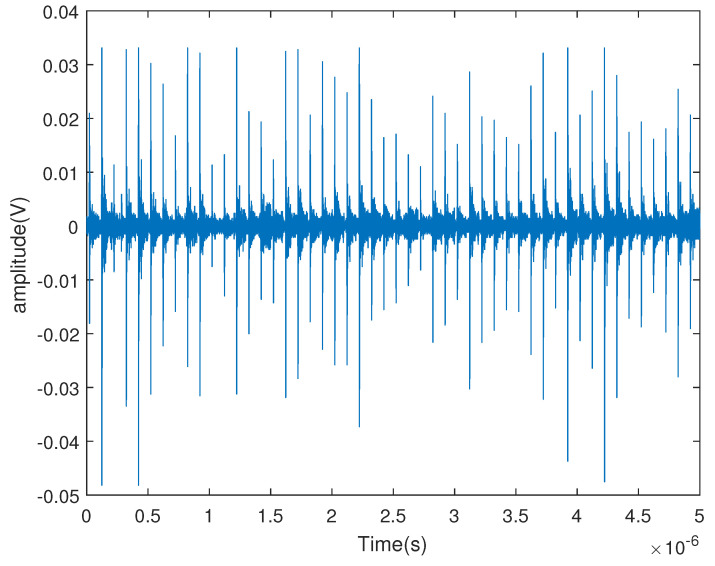
The received signal generated by the lighter.

**Figure 17 sensors-23-04562-f017:**
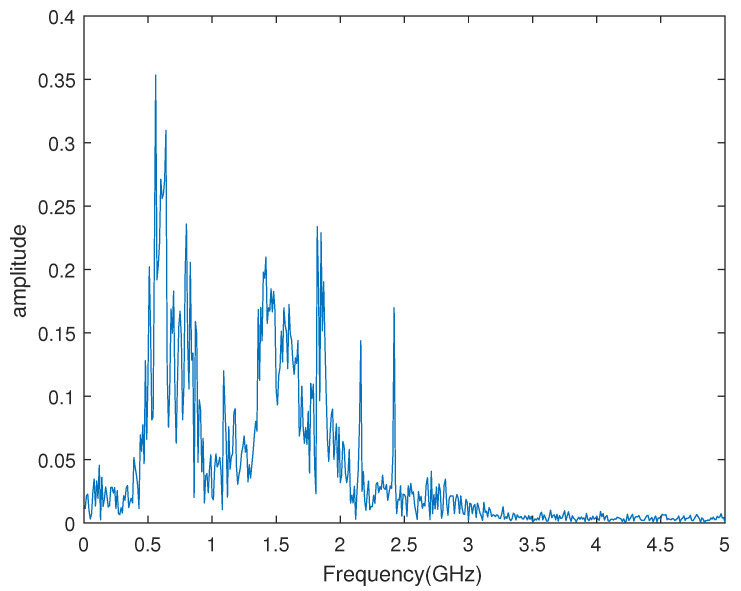
The frequency spectrum of The received signal.

**Figure 18 sensors-23-04562-f018:**
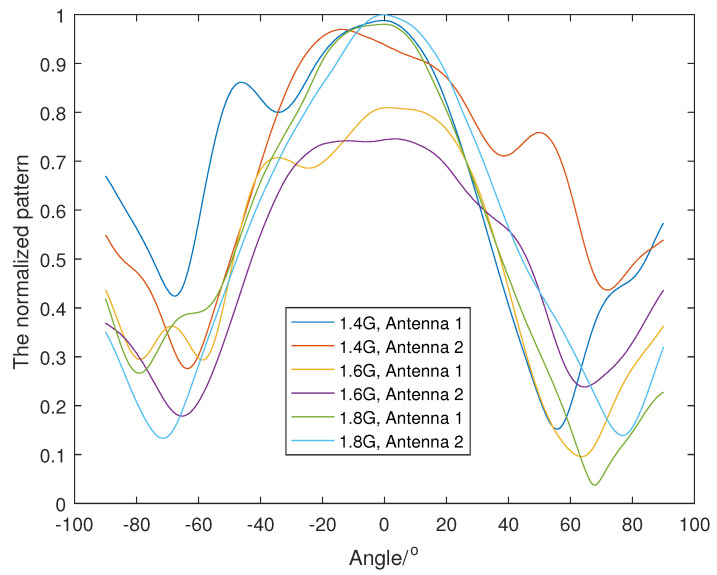
The measured antenna gain of Antenna 1 and Antenna 2.

**Figure 19 sensors-23-04562-f019:**
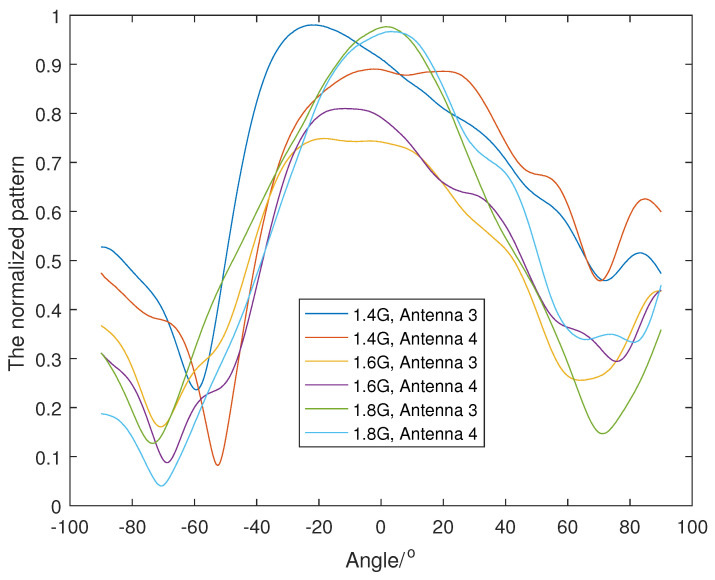
The measured antenna gain of Antenna 3 and Antenna 4.

**Table 1 sensors-23-04562-t001:** The mean errors and standard deviation of the experimental data processing.

PD Coordinates	Dir-MUSIC	ISSM	CSSM	DirSP
(7.8 m, 43°)	4.94°	−6.89°	1.88°	−0.39°
	9.10${}^{{}^{\circ}}$	3.71°	9.94°	2.96°
(7.8 m, 22°)	9.11°	3.37°	−9.65°	3.18°
	1.68°	5.28°	11.56°	2.85°
(7.8 m, 30°)	3.01°	0.11°	0.88°	−2.01°
	1.52°	4.10°	4.80°	2.98°
(14.4 m, 27.6°)	2.98°	0.38°	−1.20°	−0.12°
	2.95°	2.64°	10.02°	2.45°
(14.4 m, 39.5°)	−3.75°	−3.00°	−4.65°	−1.29°
	1.63°	2.63°	8.33°	3.46°
(6.6 m, 45.3°)	−5.37°	−10.37°	1.68°	−1.01°
	5.13°	5.12°	8.19°	2.69°

## Data Availability

All data, models, or code generated or used during the study are available from the corresponding author by request.

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
