# Peer review of "A Novel Wide-Band Directional MUSIC Algorithm Using the Strength Proportion"

_sensors, 2023, doi:10.3390/s23094562_

Round 1
Reviewer 1 Report
The authors presented A Novel Wide-band Directional MUSIC Algorithm Using the Strength Proportion. The concept is exciting, and the simulation results are reasonably good, showing strong reconfigurability. I have a minor revision before accepting it for publication:
1- The introduction needs to be improved, a total of 14 references s are not enough. Please discuss more references.
2- At the end of the paper, please compare your design with the existing work of literature.
That's all for me.
Author Response
We conduct a new experiment to verify the performance of the proposed method and have exhibited the results in the paper.
1- The introduction needs to be improved, a total of 14 references s are not enough. Please discuss more references.
Thanks for the kind suggestion, and an introduction for wideband subspace technique applications has been added. Specifically, Reference 13,15,16 and 17 have been added.
2- At the end of the paper, please compare your design with the existing work of literature.
Thanks for the kind suggestion, “Finally, experimental data processing has verified the effectiveness of DirSP. Compared to Dir-MUSIC, ISSM and CSSM, DirSP performs better both on mean errors and standard deviation. More importantly, DirSP performs more stable than other methods, while there are some bad results using other methods. Therefore, DirSP is effective for PD location.” Has been added at the end of the paper in conclusions.
Reviewer 2 Report
In this paper, the author proposes the directional multiple signal classification (Dir-MUSIC) algorithm based on the antenna gain array manifold to find the location of the Partial discharge (PD) source in substations. However, the manuscript still has the following problems:
1. How does the directional multiple signal classification (Dir-MUSIC) algorithm based on the antenna gain array manifold in this paper make practical sense?
2. In line 1, a definite article seems to be missing before the word 'antenna'. Consider adding 'the' before the word.
3. In line 19, 'is' seems to be in the wrong tense. Consider changing to 'have been'.
4. In line 21, it seems that 'substation' may not agree in number with other words in this phrase.
5. In line 148, the text's number of figures should be clickable and linked to their appropriate position.
6. In line 232, the word 'patten' doesn’t seem to fit this context. Consider replacing it with 'pattern'.
7. English writing should be enhanced considerably. There are many grammatical mistakes in the article. For example, improper use of symbols and articles, misspelling words, and so on. It is recommended to check carefully and verify corrections.
Author Response
We conduct a new experiment to verify the performance of the proposed method and have exhibited the results in the paper.
1. How does the directional multiple signal classification (Dir-MUSIC) algorithm based on the antenna gain array manifold in this paper make practical sense?
Thanks for the kind problem. The wideband Dir_MUSIC algorithm is regarded as an improvement of the Dir-MUSIC algorithm. The Dir-MUSIC algorithm can be used for PD initial location and the proposed method can be used for precise Location to find the PD source.
2. In line 1, a definite article seems to be missing before the word 'antenna'. Consider adding 'the' before the word.
Thanks for the kind suggestion, and we have modified this mistake in the paper.
3. In line 19, 'is' seems to be in the wrong tense. Consider changing to 'have been'.
Thanks for the kind suggestion, and we have modified this mistake in the paper.
4. In line 21, it seems that 'substation' may not agree in number with other words in this phrase.
Thanks for the kind suggestion, and we have modified this mistake in the paper.
5.In line 148, the text's number of figures should be clickable and linked to their appropriate position.
Thanks for the kind suggestion, and we have modified them in the paper.
6. In line 232, the word 'patten' doesn’t seem to fit this context. Consider replacing it with 'pattern'.
Thanks for the kind suggestion, and we have modified this mistake in the paper.
7. English writing should be enhanced considerably. There are many grammatical mistakes in the article. For example, improper use of symbols and articles, misspelling words, and so on. It is recommended to check carefully and verify corrections.
Thanks for the kind suggestion, we have checked for spelling and grammatical mistakes carefully in the whole paper.
Reviewer 3 Report
In this study the directional multiple signal classification (Dir-MUSIC) algorithm based on antenna gain array manifold has been proposed to find the location of the Partial discharge (PD) source in substations. In the literature one can find interst in Partial discharge (PD) detection and location systems which are indispensable for performing quality assurance and fault identification in high voltage apparatus. The presnt study propose a wideband Dir-MUSIC algorithms for PD direction finding are discussed, and a novel wideband Dir-MUSIC algorithm using the strength proportion. The study is based in a theoretical and experimental basis. It is well discussed and conducted. This study is interesting to high power substations applications. It is ready for publication.
Author Response
We conduct a new experiment to verify the performance of the proposed method and have exhibited the results in the paper.
Round 2
Reviewer 2 Report
Accept in present form